# Electroencephalogram and Physiological Responses as Affected by Slaughter Empathy in Goats

**DOI:** 10.3390/ani13061100

**Published:** 2023-03-20

**Authors:** Pavan Kumar, Ahmed Abubakar Abubakar, Muideen Adewale Ahmed, Muhammad Nizam Hayat, Mokrish Ajat, Ubedullah Kaka, Yong Meng Goh, Awis Qurni Sazili

**Affiliations:** 1Institute of Tropical Agriculture and Food Security, Universiti Putra Malaysia, Serdang 43400, Selangor, Malaysia; 2Department of Livestock Products Technology, College of Veterinary Science, Guru Angad Dev Veterinary and Animal Sciences University, Ludhiana 141004, India; 3Department of Animal Science, Faculty of Agriculture, Universiti Putra Malaysia, Serdang 43400, Selangor, Malaysia; 4Department of Veterinary Preclinical Science, Faculty of Veterinary Medicine, Universiti Putra Malaysia, Serdang 43400, Selangor, Malaysia; 5Department of Companion Animal Medicine and Surgery, Faculty of Veterinary Medicine, Universiti Putra Malaysia, Serdang 43400, Selangor, Malaysia; 6Halal Products Research Institute, Universiti Putra Malaysia, Serdang 43400, Selangor, Malaysia

**Keywords:** emotions contagion, slaughter empathy, electroencephalogram, blood glucose, animal welfare

## Abstract

**Simple Summary:**

Emotions play an important role in animal survival through better cohesion and coordination, and affect behavioral, physiological, and cognitive responses in animals. Improving positive emotions and reducing negative emotions has been advocated for better compliance with animal welfare and to improve the productivity of animals. The preslaughter handling of animals is a very crucial stage of meat production as it affects animal welfare and meat quality. The slaughter environment could lead to emotional stress in animals. There is a need to study the effect of exposure to the slaughter environment in goats.

**Abstract:**

Recent advances in emotions and cognitive science make it imperative to assess the emotional stress in goats at the time of slaughter. The present study was envisaged to study the electroencephalogram and physiological responses as affected by slaughter empathy in goats. A total of 12 goats were divided into two groups viz., E-group (goats exposed to slaughter environment, *n* = 6) and S-group (goat slaughtered in front of E-group, *n* = 6). The electroencephalogram and physiological responses in male Boer cross goats (E-group) were recorded in a slaughterhouse in two stages viz., control (C) without exposure to the slaughter of conspecifics and treatment (T) while visualizing the slaughter of conspecifics (S—slaughter group). The exposure of the goat to the slaughter of a conspecific resulted in a heightened emotional state. It caused significant alterations in neurobiological activity as recorded with the significant changes in the EEG spectrum (beta waves (*p* = 0.000491), theta waves (*p* = 0.017), and median frequency MF or F50 (*p* = 0.002)). Emotional stress was also observed to significantly increase blood glucose (*p* = 0.031) and a non-significant (*p* = 0.225) increase in heart rate in goats. Thus, slaughter empathy was observed to exert a significant effect on the electric activity of neurons in the cerebrocortical area of the brain and an increase in blood glucose content.

## 1. Introduction

Emotions are very intense, short-term positive or negative state responses to external or internal stimuli of specific importance for a living being. Emotions determine the behavioral decisions to approach or avoid stimuli [1]. Emotions allow animals to cope with situations with negative or positive meanings and involve certain neurophysiological responses [2]. Emotions play an important role in animal welfare and it has been advocated to improve positive emotions and reduce negative emotions for better compliance with animal welfare and to improve the productivity of animals [3]. This area is getting due recognition in the last two decades with a focus limited to pharmaceutical applications and animal welfare compliance by studying animal behavior (Ethology). Analyzing and comprehending the emotional experiences of an animal could provide information about its welfare status [2]. The present research focus in ethology is to apply innovative research frameworks such as studying the valence (activation) and arousal (excitation) aspects of emotions [4] with various indicators such as neurophysiological indicators (heart rate, brain activity, neuroendocrine response) [5,6], behavior indicators (facial expression, vocalization, tail, and ear postures) [5,7], facial expressions [2,8], and cognitive changes (judgment biases) [9]. 

The emotional changes (vocalization and facial expression) can be detected by conspecifics through olfactory, visual, or audible means, and an automatic trigger state matching between two individuals (emotional contagion) [10]. In the whole process, one animal is affected and shares the emotions of another conspecific via empathetic processes [11]. This emotional contagion helps in regulating social interactions and the fast exchange of information among group members. It facilitates better cohesion and coordination among group members in defense (in case of negative emotions such as fear due to the presence of a predator) or better group adhesion in positive emotions [12]. This emotion sharing leads to cognitive forms of empathy comprising sympathetic and empathetic concerns in turn helping the receiver to downregulate its own emotional response by effective sharing among conspecifics [13]. The sympathetic form of empathy could be widespread among animals but still lacks proper information in non-human animals due to a lack of a suitable methodology or experimental design [13,14]. 

Goats are small ruminants that contribute significantly to the socio-economic development of rural economies owing to their survival and productivity in a harsh climate, disease resistance, low neophobic responses, ability to cope with stressors, and inquisitiveness [15,16,17]. Goats have the capability of identifying and responding to calls with different emotional valences such as food frustration or reward and were observed to have head-orientation bias to the right side upon the vocalization of a conspecific in the context of frustration and dog barks indicating frustration [18]. Similarly, positive and negative emotional-linked vocalization was reported to affect the behavior and cardiac response in goats [19]. 

There are several reports (published reports, spy cameras, hidden videos, etc.) mentioning the improper handling of animals during slaughter [20,21,22,23,24,25]. However, physical mishandling is widely reported and studied in the slaughter of animals, but the emotional mishandling of animals during slaughter has been largely overlooked. Positive emotions could promote positive welfare among livestock [19]. During slaughter, animals undergo severe emotional stress and distress due to the slaughterhouse environment (novelty, noise, unfamiliar animals, objects, and persons), the odor of blood and animal waste, and animals’ vocalization. Such types of situations are widely prevalent in both developing and developed worlds [18,19,20,21,22,23]. Further, at the time of religious sacrifice of animals during festivals, animals are slaughtered in groups in front of conspecifics. Research on emotional stress as affected by slaughter empathy is scarce. With the advancement in cognitive science in non-human animals, it is becoming imperative to study this aspect of non-human animals with an appropriate research methodology [26]. 

Recently, electroencephalogram (EEG) has increasingly been used for assessing pain and stress during the slaughter of livestock [27]. It is a technology used to measure the electric activity of neurons in the cerebrocortical region of the brain by fixing electrodes on various positions of the brain [28,29]. The electric activity of neurons was used to assess pain and stress in animal welfare during the slaughter process in goats [30,31,32,33]. However, the high cost of equipment, experimental conditions, and analysis still remain issues in its popularization. To the best of our knowledge, there is no published study available on the application of EEG in assessing potential pain and stress during exposing animals to a slaughter environment. The neural oscillation/electric signals produced by the cortical pyramidal neurons upon various emotions or feelings could be measured by placing electrodes at different areas of the scalp and these signals could be analyzed by various EEG spectrum variables such as frequencies, timings, total energy, and amplitudes [34]. 

Thus, the present study was designed to evaluate the oscillation/electric signals produced by the cortical pyramidal neurons via the EEG recording during slaughter empathy in goats. The physiological parameters were also assessed to correlate these with the EEG power spectrum. 

## 2. Materials and Methods

### 2.1. Ethical Approval

The present study was conducted following the animal ethics guidelines of the Research Policy of Universiti Putra Malaysia as per Institutional Animals Care and Use Committee approval No.: UPM/IACUC/AUP-R003/2022, Dated 27 May 2022.

### 2.2. Animals

Goats (12 Boer cross, age 12 months, 25–30 kg live weight) were purchased from the local market (Global Field Trading, No 12, Jalan 9/6, Seksyen 9, 43650 Bandar Baru, Bangi, Malaysia). These animals were housed at a small ruminant housing facility at the Institute of Tropical Agriculture and Food Security (ITAFoS) in Universiti Putra Malaysia, located on latitude 259′06.5″ N and longitude 101^0^43′40.7″ E (Jalan Maklumat) for 14 days adaptation period. Animals were housed separately with 0.3 m^2^/animal size individually in naturally ventilated pens. During the stay, animals were fed twice daily and accessed the ad libitum freshwater source. Animals had proper access to veterinary services, and various physiological parameters were recorded daily on the animal monitoring sheet (heart rate, rectal temperature, breathing rate, normal/abnormal movement, and normal/abnormal activity). Prior to the start of the experiment, the animals found suitable were transported (2.0 km) from the farm to the research slaughterhouse of the Department of Animal Science, Faculty of Agriculture, Universiti Putra Malaysia (258,059.000 N; 10,144,006.400 E). The animals were rested overnight in the lairage with ad libitum drinking water availability. A trained veterinarian conducted the ante-mortem and post-mortem inspection during the slaughtering process. 

### 2.3. Experimental Conditions and Design

This study was conducted in September and October 2022. Animals were assigned into two groups with the treatment group exposed to the slaughter environment and emotional stress (*n* = 6) (E group) while slaughtering the other animal (S-group). The goats were Halal slaughtered by transverse severance of the carotid arteries and jugular veins as per the standard protocols outlined in the MS 1500:2009 (Department of Standards Malaysia, 2009). Figure 1 presents the experimental design of the experiment. 

### 2.4. Electroencephalogram Recording

The EEG sampling of goats was performed at the slaughter hall. The goat (treatment group) was moved from the lairage to point of slaughter by using a race and EEG sampling was performed in the absence of another goat. After that, another animal (from S-group) was taken to the slaughter point and slaughtered. To mimic the normal practice following slaughter in developing countries and in religious sacrifice, the goats were exposed to slaughter environment by auditory, olfactory, and optic senses. The distance between the goat visualizing the slaughtering and undergoing the emotional stress of slaughtering environment and the slaughtered goat was kept at approx. 2 m throughout the study. Goats were standing and restrained minimally during the whole process of the EEG reading. The whole slaughter process was completed within 4–6 min. After each slaughter, the site was thoroughly washed with water before repeating the same process for another pair of animals. 

The EEG sampling was performed by using two conductive electrode patches attached to the zygomatic process of the frontal bone and the mastoid area by following the method as followed by Sabow et al. [30]. A fur area of 5–6 cm diameter was shaved (5–6 h prior to study) in between the mastoid process and the medial canthi of the eyes. The area was cleaned and gently rubbed with cotton rolls containing 70% ethanol to degrease the area, thereby improving the attachment of electrode gels hydrogel conductive adhesive sterile disposable electrodes (Covidien LLC, Mansfield, MA, USA) to the shaved skin. It was ensured to properly shave and clean the area to improve the quality of signals, thereby EEG quality. A negative (inverting) electrode was also placed on the zygomatic process of the frontal bone (on the right side, 1.5–2.0 cm below eye level). The positive (non-inverting) electrode was placed on the cleaned mastoid process [35]. The attachment of electrodes is depicted in Figure 2. 

The EEG recording was carried out by using Powerlab 4/20 data recording system (Powerlab data acquisition system, ADInstruments Ltd. Sydney, Australia) with the help of Chart 5.0 (PowerlabTM data acquisition system, Sydney, Australia) installed in a laptop. The EEG recording was started within 30 s upon the placement of the electrodes and recorded for 5–7 min on E-group goat till another goat undergoing slaughtering from S-group was dead. The determination of the state of death was confirmed by the absence of pupillary and corneal reflexes, flaccid tongue, absence of breathing, and fully dilated pupils as per Malaysian Protocol for the Halal Meat and Poultry Production, Department of Islamic Development, Malaysia (MS 1500:2009). 

Observations were made during the EEG reading to record the artifacts resulting from the physiological rhythmic movements such as eyelids or cardiovascular movements, electrical interferences, and physical movements from the goats themselves, such as ear flapping or rumination. EEG activities were analyzed later offline using the Chart 5.0 software (ADInstruments Ltd., Sydney, Australia). The EEG was recorded at a sampling rate of 1 kHz. The individual power spectrum of alpha, beta, delta, and theta waves was calculated based on the amplitude and frequency of the EEG signals [33]. Artifacts were removed from the overall activity, and individual waves were subjected to fast Fourier transformation (FFT) analysis. FFT is a mathematic tool that assists in quantifying information within the raw EEG signal by changing the raw EEG signal to the frequency domain from the time domain, thereby generating a power spectrum. Total power (Ptot, the total area under the curve), root mean square (RMS), and median Frequency (F50, frequency below which 50% of Ptot lies) were calculated repeatedly for non-overlapping of one-second epochs, yielding 60 epochs per minute [35]. A 60-s block EEG data were collected in E-goat at the control value (prior to exposure to the slaughtering process) and after 90 s of neck cut of S goats (treatment value). Each block was calculated for consecutive non-overlapping 1-s epochs. 

The EEG power spectrum is depicted in Figure 3a,b. 

Figure 3a,b represent the electroencephalogram’s electrical activity categorized as delta (4 Hz), theta (4–7 Hz), alpha (8–13 Hz), or beta (>13 Hz) waves. Figure 3a refers to the EEG power spectrum recorded in E-goat during the control state without exposure to slaughter environment. Figure 3b refers to the EEG power spectrum recorded in E-goat during the treatment phase by exposing to slaughter environment by slaughtering a goat from the S-group. A black box on the screen represents an epoch length.

### 2.5. Physiological Responses 

The physiological responses to emotional stress during exposure to the slaughter process in goats (E-group) were assessed by measuring heart rate (by stethoscope), rectal temperature (by thermometer), and blood glucose (by portable blood glucometer by putting a drop of blood on test strip onto the device) before bringing the S-goat (control phase) and after 1.5–2 min of neck cut of S-goat (treatment phase). The animal was restrained minimally during the whole process and heart rate and blood collection were undertaken by experienced technical staff by gently placing knees behind the shoulder and 30^0^ raising the animal head at lairage and immediately after exposure.

### 2.6. Statistical Analysis

The data were tested for normal distribution using a Shapiro–Wilk test using SPSS Statistics Version 20 software (IBM Corporation, New York, NY, USA). Paired *T*-test was used to determine the differences in values between pre-T and post-T (*n* = 6). A *p*-value of less than 0.05 was considered statistically significant.

## 3. Results and Discussion

### 3.1. EEG Variable

The emotional stress due to exposure to a slaughter environment in goats was observed to have a significant effect on the electrical activity of neurons as recorded by the EEG power spectrum (Table 1).

An in-depth analysis of the EEG power spectrum could provide details about the changes in the electrical activity of cerebrocortical activity [33]. These neurons are widely acknowledged to play an important role in pain perception [36]. An increase in the brain activity of conscious animals during the preslaughter handling and slaughter was proposed to be associated with pain sensation [37].

#### 3.1.1. Alpha, Beta, Gamma, and Theta Waves Pattern

The alpha waves of the goats were observed to have a non-significant increase (*p* > 0.05) during the treatment phase (exposure to the slaughter process/slaughter environment) as compared with the control phase. Alpha waves have a frequency width of 8–12 Hz and, in humans, these waves are correlated with auditory and visual stimulations with memory-related events [38]. Various slaughter environments could be attributed to the increase in the alpha waves. The alpha waves were also recorded as significantly higher in lairage compared with the baseline value after 6 h of transportation in goats [31]. An increase in alpha wave activity was also recorded in lambs and goats after head-only and head-to-back electrical stunning [39]. 

The beta waves significantly (*p* = 0.000491) increased due to treatment compared with the control value. The beta waves in the EEG power spectrum increased upon the increased brain activity [40]. The beta waves were reported to increase during brain activity in a panic condition [38,41]. In humans, a higher beta power was recorded under stressful conditions [42]. Several studies have recorded an increase in the beta waves during stress in animals such as during the transportation and slaughter of goats [31,32]. 

The delta waves followed a similar trend to that of the alpha waves with the treatment phase having higher but comparable values to the control phase. Higher delta waves were recorded in goats during slaughter compared with the corresponding value at the farm [31]. These waves are associated with a brain’s default mode network [43]. The theta waves in the treatment phase were recorded as significantly (*p* = 0.017) higher than their control values. In humans, these waves represent a heightened emotional state with increased alertness and arousal [38,41]. The theta waves were also recorded significantly higher under pre-slaughter stress but not under transport stress [31]. In horses, the increased theta wave was correlated with stereotypic behavioral performance/compromised animal welfare, and horses with a good welfare status had lower gamma waves in the right hemisphere [44]. The significant increase in the theta waves in the present study could be due to the heightened emotional status of goats arising due to exposure to the act of slaughter. 

Similarly in humans, Kim et al. [45] proposed the accurate and early detection of emotional stress and stages of stress by recording EEG using three-dimensional (3-D) convolutional neural networks by considering theta, alpha, beta, and gamma wave patterns. Further, a comparative value of theta/beta power was reported to be used in the stress monitoring system with more than 90% accuracy and classifying stress in a low level, a moderate level, and a high level [46]. 

In our present study, a significant change occurred in the value of beta and theta waves. Sabow et al. [33] observed an increase in brain activity of goats as reflected by the EEG power spectrum due to stress during slaughter. Similarly, Zulkifli et al. [47] observed the changes in alpha, beta, theta, and delta waves of the EEG power spectrum under different stunning and slaughter methods in cattle. 

#### 3.1.2. Ptot and F50

The total power of the EEG spectrum (Ptot) (Cohen’s d value- −0.34) showed a non-significant increase during the treatment phase as compared with the control phase (Figure 4). The Ptot of the EEG spectrum correlates with the relaxed phase of animals, with animals in the relaxed phase usually having lower power [27]. The median frequency (MF or F50) of the EEG spectrum was recorded as significantly (*p* = 0.002) higher in the treatment phase compared with the control. An increase in the median frequency in the EEG spectrum typically indicates stress or painful conditions [35]. 

Under situations of pain, the EEG spectrum was noticed to have a significant (*p* < 0.01) effect on Ptot, F50, and F95 (95% spectral edge frequency) values in lamb during the castration process [48]. Under a conscious state, EEG could be used as a tool to assess pain and stress in animals by measuring F50 and F95 [48]. The increase in the F50 of the EEG power spectrum was related to noxious stimulation and pain during the neck cut [34]. During slaughter, ventral neck cuts in goats were observed to have a significant increase on the F50 as compared with the stay in lairage [31]. 

Reports regarding the changes in Ptot in association with F50 are inconsistent. Murrel and Johnson [49] observed a decrease in Ptot with an increase in F50. Imlan et al. [50,51] and Abubakar et al. [52] in cattle and Raghazli et al. [31] in goats also reported a positive correlation in Ptot and F50 under stress as well as noxious stimuli. On the other hand, Kaka et al. [29] and Karna et al. [53] reported no change in Ptot in association with F50 in response to noxious stimuli under anesthesia. Thus, the results of this study also show a trend similar to that reported by Kaka et al. [29] and Karna et al. [53] in dogs under anesthesia, however, the present study was conducted in conscious goats. It has been reported that changes in the Ptot were not directly associated with F50 in response to noxious stimuli [29] and that these changes could represent a different component of nociception than F50 [53]. These lines of evidence, including the present study, verifies that Ptot is not directly associated with MF and might have a connection with other components of pain and stress, which is yet to be explored. 

Thus, the higher F50 in the present study (Cohen’s *d* value- 0.79) could be correlated with emotional stress in goats. The higher responses of F50 and F95 were reported to associate with pain in calves [54,55,56]. The EEG spectrum was also noticed with increased F50 and Ptot [33]. Furthermore, in gregarious animals such as goats, the visual and physical separation from their herd and the novelty of the environment could be an additional factor that contributed to substantial fear and anxiety at the slaughter point in the present study [57]. 

Several psychological studies on human subjects have also confirmed the long-lasting effect on stress hormones due to negative emotions such as anger, disgust, and fear, whereas positive emotions help in the production of beneficial hormones [45]. The EEG has been used in humans for emotion recognition with remarkable results [58,59,60]. However, there is a lack of studies on emotional stress in animals by applying EEG recording. To the best of our knowledge, this is the first study recording emotional stress and slaughter empathy in goats upon exposure to the act of slaughter by measuring the electric activity of cerebrocortical neurons. 

During slaughter, the overall slaughter environment is the key determinant in affecting animals’ physiological and emotional states. The emotional response in animals comprises behavioral, physiological, cognitive, and subjective components [61]. This emotional stress resulted in changing the EEG variables and physiological responses in the goats. As emotional stress is of a very short duration in the context of slaughter, with varying degrees of intensity or threat levels, measuring these slaughter empathy reactions or various neurobiological responses warrants using an appropriate methodology that records these variations instantaneously, sensitively, and accurately. EEG could be used in non-human animals to recognize emotions up to the point of slaughter [62,63,64].

### 3.2. Physiological Responses

The stimuli at the slaughterhouse differ from the farm, and this may affect the emotional status of animals, and their transport from the farm to the slaughterhouse may further aggravate it [65]. Goats, being a prey animal, have well-developed mechanisms to respond to any situation or potential state of threat or danger. The physiological responses in goats were recorded with higher values for heart rate (*p* = 0.225) and a significant increase in blood glucose (*p* = 0.012) during the stage of exposure to the slaughter of conspecifics (Table 2). However, the temperature (both rectal and auditory) was recorded as comparable in the present study. 

Semiochemicals released from the blood of the slaughtered animals were considered a major factor in causing distress in animals during slaughter. As per Grandin and Vogel [66], the vision or smell of blood is not thought to cause distress unless the animal whose blood is present had been distressed during slaughter (e.g., he or she struggled and vocalized). Preslaughter handling had been established to affect animal welfare and meat quality [67,68]. 

An increase in the glucose value in a conscious state is regarded as an indicator of stress in goats [69]. Various stress factors affect the heart rate and blood glucose levels in goats due to the increased release of catecholamines and glucocorticoids. It facilitates increasing glucose production from glycogenolysis and gluconeogenesis required for preparing animals for the response to a stressor (fight or flight response) [67,70]. An increase in blood glucose concentration was also reported during stressful conditions (pasture and slaughter) in deer [71]. Sim et al. [72] recorded increased blood glucose in mice upon emotional stress via the activation of adrenergic and glucocorticoid responses. 

In the present study, the blood glucose levels were recorded within the normal range of glucose in animals (4.4–6.6 mMol/L) [73,74]. Further, the increase in heart rate could be correlated with the various frequency bands of EEG in sheep during slaughter [75]. 

## 4. Conclusions

Based on the present study, it can be concluded that the exposure of goats to the slaughter of conspecifics alters the emotional state of goats, consequently causing significant changes in neurobiological activity as recorded with the significant changes in the EEG spectrum (beta waves, theta waves, and MF 50). Emotional stress was also observed to significantly increase blood glucose levels with no difference (*p* = 0.225) in the heart rate in goats.

## 5. Limitation and Future Direction

This study highlighted the issue of emotional stress in goats upon exposure to the slaughter environment. There is a need to take further studies on evaluating its effect on other common and established stress biomarkers such as stress hormones. Its overall impact on meat proteomics and meat quality should be assessed. 

As in the present study, all three senses (auditory, olfactory, and optic) were used to mimic the common practice in some places. A further study on which sense has more/no effect could be useful, so to provide accurate and practical recommendations for such situations. 

There are some practical challenges while conducting such studies due to potential ethical issues as the present study correlated the higher electric activity cerebrocortical neurons and physiological parameters with the emotional stress during the slaughter of goats. There is a need to take more in-depth studies to confirm slaughter empathy, which if established could have wide implications in the goat meat industry such as the requirement of slaughter out of sight of a conspecific. Interestingly, the same is recommended for Halal slaughter management. 

Such data will be valuable in increasing awareness among common people and also sensitizing the person involved in the meat industry, thereby improving animal welfare standards. However, there is a need to take a study on emotional stress and slaughter empathy in goats with a higher sample size to get more insights into this aspect. 

## Figures and Tables

**Figure 1 animals-13-01100-f001:**
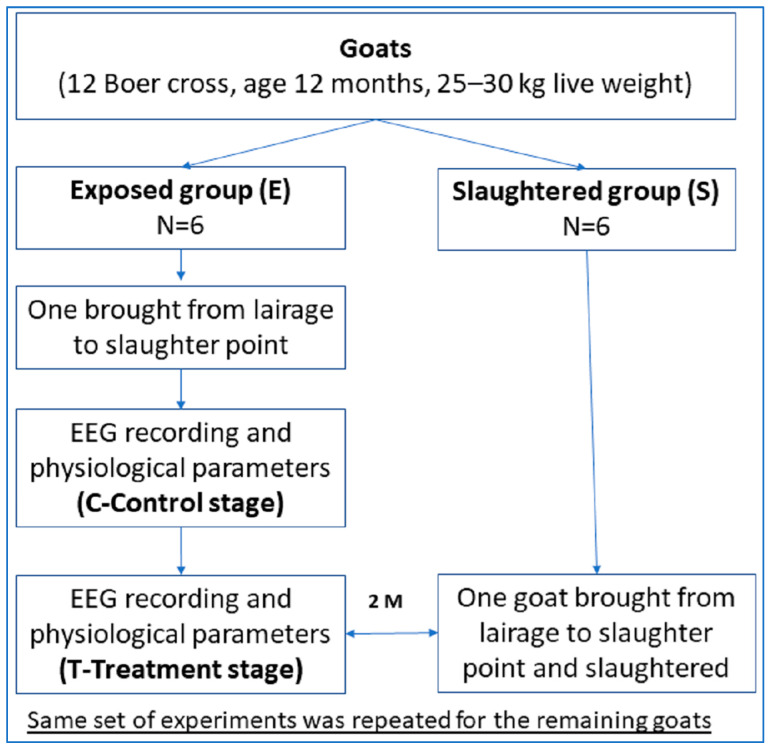
Experimental design of the experiment.

**Figure 2 animals-13-01100-f002:**
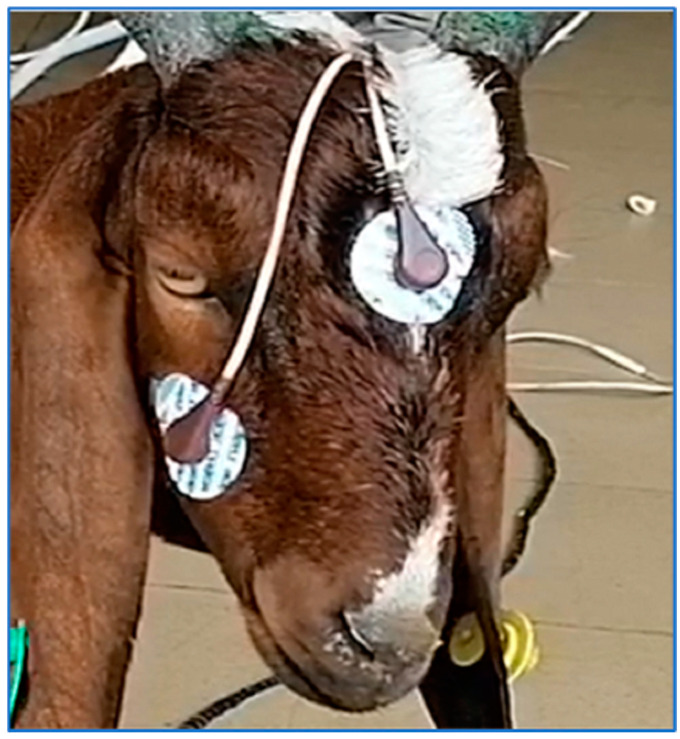
Placement of the EEG electrodes.

**Figure 3 animals-13-01100-f003:**
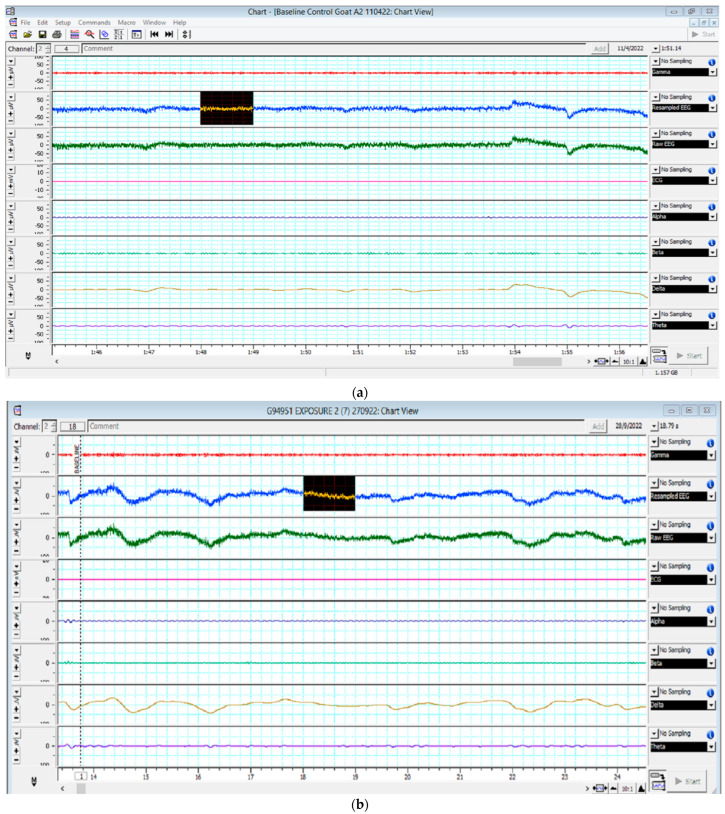
(**a**): EEG power spectrum during control state. (**b**): EEG power spectrum during treatment state.

**Figure 4 animals-13-01100-f004:**
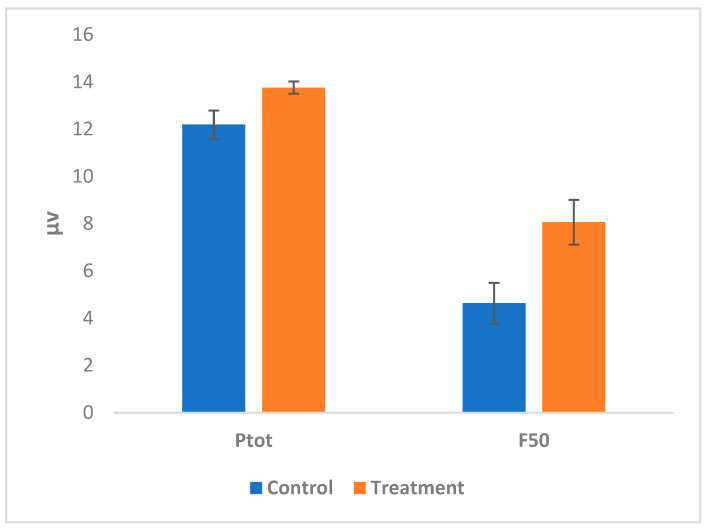
Differences in Ptot and F50 electroencephalogram RMS in goats upon exposure to act of slaughter. (Ptot—total power of EEG spectrum; F50—median frequency of EEG spectrum, Control—state without exposure to act of slaughter, Treatment—state during the exposure to act of slaughter).

**Table 1 animals-13-01100-t001:** Differences in electroencephalogram variables of goats upon exposure to act of slaughter.

Parameters	Control Stage	Treatment Stage	*t* Value	*p*-Value	Cohen’s *d* Value
Alpha (µv)	0.999 ± 0.062	1.152 ± 0.088	−1.490	0.144	−0.32
Beta (µv)	1.402 ± 0.049	1.832 ± 0.098	−3.818	0.000 *	−0.89
Delta (µv)	9.050 ± 0.814	9.693 ± 0.812	−0.521	0.605	−0.13
Theta (µv)	1.741 ± 0.114	2.266 ± 0.215	−2.508	0.017	−0.49

Values are mean ± standard error, RMS—root means square, Control—state without exposure to act of slaughter, Treatment—state during the exposure to act of slaughter, * 0.000491. Cohen’s *d* value of 0.2, 0.5, and 0.8 indicates small, medium, and large effect, respectively.

**Table 2 animals-13-01100-t002:** Differences in heart rate, body temperature, and glucose in goats upon exposure to act of slaughter.

Parameters	Control Stage	Treatment Stage	*t* Value	*p*-Value	Cohen’s *d* Value
Heart rate (beats/min)	79.33 ± 9.03	88.17 ± 12.96	−1.385	0.225	−0.91
Temperature (rectal, °C)	37.23 ± 0.26	37.33 ± 0.16	−0.278	0.790	−0.18
Temp (auditory, °C)	38.83 ± 0.17	38.99 ± 0.24	−0.855	0.425	−0.29
Glucose (mMol/L)	4.12 ± 0.16	4.88 ± 0.23	−2.969	0.031	−1.65

Values are mean ± standard error, Control—state without exposure to act of slaughter, Treatment—state during the exposure to act of slaughter., Cohen’s *d* value of 0.2, 0.5, and 0.8 indicate small, medium, and large effect, respectively.

## Data Availability

The data presented in the study are available on request from the corresponding authors.

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
