# Peer review of "Electroencephalogram and Physiological Responses as Affected by Slaughter Empathy in Goats"

_animals, 2023, doi:10.3390/ani13061100_

Round 1

Reviewer 1 Report

This is a well written and scientifically interesting paper that covers the quality criteria of ANIMALS. 

Please delete lines 93-94, and correct 'two stages viz.' with the appropriate term.  

Author Response

The authors are thankful to the anonymous reviewer for his critical comments and observations. These comments have helped us in improving the quality of the manuscript. We have edited the manuscript accordingly. Further, it is certified that all the issues raised by the reviewer have been incorporated into the revised manuscript. All the changes were marked in RED color text.

Comment: This is a well-written and scientifically interesting paper that covers the quality criteria of ANIMALS.

Response: Thank you so much for your encouraging and positive observation.

Comment: Please delete lines 93-94, and correct 'two stages viz.' with the appropriate term.

Response: Deleted

Reviewer 2 Report

The methodology and research design should be improved. The experimental conditions and circumstances are not clear enough.

Presenting of the results needs clarification in some points.

All comments and suggestions of the reviewer can be found in the attachment.

Author Response

The authors are thankful to the anonymous reviewer for his critical comments and observations. These comments have helped us in improving the quality of the manuscript. We have edited the manuscript accordingly. Further, it is certified that all the issues raised by the reviewer have been incorporated into the revised manuscript. All the changes were marked in RED color text.

General comment: The methodology and research design should be improved. The experimental conditions and circumstances are not clear enough. Presenting of the results needs clarification in some points.

Response: The methodology and research section have been edited with more clarity on the experimental conditions by adding Flowchart (Fig 1).

Comment: L22-23: animal number and group

Response: The experimental design was clarified as -A total of 12 goats were divided into two groups: E-group (goats exposed to slaughter ambient) and S group (goat slaughtered in front of E group). The electroencephalogram and physiological responses in six male Boer cross goats (E-group) were recorded in a slaughterhouse in two stages viz., control (C) without exposure to the slaughter of conspecific and treatment (T) while visualizing the slaughter of conspecific (S-slaughter group). We have added Fig 1 (Flowchart) for a better understanding of the experimental design.

Comment: L45-45-These items were not examined in the study. Why? These elements would have been easily observed and described.

Response: Authors appreciate the observation made by the reviewer. These behavioral parameters were also recorded and at present are undergoing evaluations.

Comment L68: In which parts of the world/countries? In the EU, there are very strict regulations (and sanctions, as well) for this.

Response: Authors are in agreement with the statement and there are strict regulations are placed worldwide for proper animal welfare compliance. However, as per some reports, these unacceptable practices are still continuing due to lack of awareness, job stress, etc. There have been reports of mishandling in Bangladesh  (Ahsan et al 2014), US and other countries (Grandin et al., 2014), Chile (Gallo et al., 2018), UK (Gregory et al., 2009), Ghana (Firmpong et al., 2014) 

Comment: L75-76: reference

Response: reference added  [18-23]

Comment: L78: are-change to is

Response: Corrected

Comment: L103: )

Response: ) Added

Comment: ad libitum in italics

Response: Corrected

Comment: It is not clear enough. In Abstract section, you mention C and T groups (n=6 in each, as I understand). Here, you mention S group without C and T. Please clarify which group was slaughtered, which group was treated and which group was the control.

Response: Thank you for pointing this. There were 12 goats used in the study divided into two groups viz. animal exposed to the slaughter process (E-group) while slaughtering the other animal (S-group).

Comment: How long did it take per animal? How much time passed between this process and the slaughtering? This procedure may have caused stressful effects to the animals, increasing heart rate and/or cortisol levels.

Response: The shaving of fur was done well before the start of the experiment (5-6h prior to the study). The EEG recordings were taken within 30 s of placing conductive electrodes on the goat (E group). After 60 s recording, a goat (S group) was brought and slaughtered which took 4-6 min for completing the whole slaughtering process.

Comment: How were the electrodes stay on the animals? If they were allowed to move, I'm sure they tried to remove them by shaking their heads. And if they were tied, then additional stress and animal welfare issues arise.

Response: The electrodes are attached via adhesive hydrogels gently. Yes we agree as in animal nature, there were some little head movements leading to artifacts. During analysis, artifacts were removed from the overall analysis.

Comment: L174: S group

Response: S group- goat slaughtered for exposing to slaughter to E-group goat.

Comment: Both figures must be explained.

Response: The following note is added -

[Fig 2a and 2b represent the electroencephalogram’s electrical activity categorized as delta (4 Hz), theta (4–7 Hz), alpha (8–13 Hz), or beta (>13 Hz) waves. Fig 2a refers to the EEG power spectrum recorded in E-goat during the control state without exposure to slaughter ambient. Fig 2b refers to the EEG power spectrum recorded in E-goat during the treatment phase by exposing to slaughter ambient by slaughtering a goat from S-group.]

Comment: Detailed description needed. All of these treatments can cause stress to the animals. How did you eliminate these effects?

Response: A detailed description on measuring these parameters was added. The animal was handled minimally by experienced staff and trained technical staff measured heart rate, temperature, and blood to minimize the stress.

Comment: How were the blood tests carried out?

Response: Added as- A blood glucose test was conducted by a portable blood glucose meter-glucometer by putting a drop of blood on a test strip into the device

Comment: The treatment phase

Response: stage of recording various parameters while exposing to slaughter ambient.

Comment: Later you cite reference No. 51. What did Sabow et al. (2019) do differently?

Response: Sabow et al measured the EEG variables and blood parameters in anesthetized goats by subjecting slaughter without stunning vs slaughter with electrical stunning. In Sabow et al. study there was no exposure to slaughter ambient. Our present study measured the EEG and physiological responses in goats while exposing to slaughtering ambient.   

Comment: L258: HERE

Response: Authors are sorry for the typo error as it was mentioned to indicate the place of Figures. But while the conversion of word to MDPI format, it remained as such.

Reviewer 3 Report

This manuscript details an EEG and physiological study of goats watching another goat get slaughtered. It is a useful addition to the literature, and more research is needed about the impact of watching slaughter on animals, as this is probably common practice in many parts of the world. Before I can recommend it for publication, I would like the authors to consider some modifications.

Abstract and results - could the authors present effect sizes for their analyses? That would tell us whether the differences observed have any meaning in the real world.

L57-59 this phrasing is unclear. Please rewrite. How can something be widespread but with a lack of information? If there's not enough information, how can we know it's widespread?

L65 - what is the implication of this right head turn bias? What does that indicate?

L72-75 - please cite this claim and the claim in the next sentence (L75-77). These are very important because they are necessary to justify the study. This is an ethically difficult study, because it's knowingly subjecting an animal to an event that is potentially very distressing. In order to make it ethically acceptable, on the basis that it is happening all the time in real life anyway so we need to know the impact on the animals, it is very important to cite these claims.

L89 - how is this study different from the current study? Both look at EEG in goats during slaughter. There's no noted gap in the research that leads to the current study. Please add a sentence explaining why the current study still needs to be done.

L125 - so there was no condition/sampling period when the goat was with a conspecific before the slaughter experience? The 'control value' was mentioned in L173 but it is unclear whether another animal was present at that point. If so, this is a major limitation, because it's impossible to know whether the mere presence of the other animal was causing the EEG/physiological reaction that the goat was experiencing. Please clarify if this was incorrect. If it is the case, it might be a dealbreaker because there's no true baseline.

L154 - was it 5-7 minutes or 4-6 mins at noted on L133?

L173 - how much of each goat's recording was excluded due to artifact? How much usable data was obtained for each?

Fig 2 - what do the black boxes on the screen represent? An epoch length? If so, add to caption/note on the figure.

Results and Discussion

I suggest moving L193-209 below the results, because they are more general statements. It would make more sense to read the results and their interpreting statements first, and then go to the more general context.

L211-215 - suggest moving below Table 1 for the same reason.

Table 1 - what does RMS stand for in the caption?

Table 1 - the note about superscripts is unnecessary because that is obvious from the p-values, since there are only two groups. Usually the superscripts are only necessary when 3+ groups are being compared.

L238 - what do the delta waves represent? The others were all explained but not this one.

L245 - why were the gamma waves not reported for this sample? Especially when they are discussed in the Discussion as being important?

L247 - I would avoid using the term 'slaughter empathy' because we don't know that this is what's going on. It looks like the animals are experiencing high levels of stress, but this could be emotional contagion, a cousin of empathy which is well-established in animals, but is not believed to be a type of cognitive empathy, as slaughter empathy would be.

L258 - 'HERE' is at the end of this sentence.

L330 - where are the limitations and future directions? What about the practical applications? Just saying that future research with a larger sample needs to be done isn't sufficient. There are other limitations that should be noted, and there are plenty of future directions that could be mentioned. Also, what does it mean for real life goats in livestock industries? Should they be protected from witnessing slaughter of other goats? Is this realistic?

Author Response

The authors are thankful to the anonymous reviewer for his critical comments and observations. These comments have helped us in improving the quality of the manuscript. We have edited the manuscript accordingly. Further, it is certified that all the issues raised by the reviewer have been incorporated into the revised manuscript. All the changes were marked in RED color text.

General comment: This manuscript details an EEG and physiological study of goats watching another goat get slaughtered. It is a useful addition to the literature, and more research is needed about the impact of watching slaughter on animals, as this is probably common practice in many parts of the world. Before I can recommend it for publication, I would like the authors to consider some modifications

Response: Thank you so much for your positive and encouraging observations. It means a lot for us.

Comment: Abstract and results - could the authors present effect sizes for their analyses? That would tell us whether the differences observed have any meaning in the real world.

Response: Authors fully agreed with the issue of effect size pointed out by the anonymous reviewer. The effect size in terms of Cohen’s d value was calculated by using an effect size calculator by the University of Colorado USA  (https://lbecker.uccs.edu/) for each parameter and presented in the tables. Further, a description of these values was added in the footnote for each Table.

Comment: L57-59 this phrasing is unclear. Please rewrite. How can something be widespread but with a lack of information? If there's not enough information, how can we know it's widespread?

Response: Sentence edited to could be widespread

Comment: L65 - what is the implication of this right-head turn bias? What does that indicate?

Response: Sentence edited by adding as indicating frustration.

Comment: L72-75 - please cite this claim and the claim in the next sentence.

Response: References 18-23 cited.

Comment: L75-77). These are very important because they are necessary to justify the study. This is an ethically difficult study, because it's knowingly subjecting an animal to an event that is potentially very distressing. In order to make it ethically acceptable, on the basis that it is happening all the time in real life anyway so we need to know the impact on the animals, it is very important to cite these claims.

Response: Authors are very thankful for the points mentioned by the reviewer; these would be very helpful in improving the hypothesis and included in the revised draft.

Comment: L89 - how is this study different from the current study? Both look at EEG in goats during slaughter. There's no noted gap in the research that leads to the current study. Please add a sentence explaining why the current study still needs to be done.

Response: Added the gap- To the best of our knowledge, there is no published study available on the application of EEG in assessing potential pain and stress during exposing animals to slaughter ambient.

Comment: L125 - so there was no condition/sampling period when the goat was with a conspecific before the slaughter experience? The control value' was mentioned in L173 but it is unclear whether another animal was present at that point. If so, this is a major limitation, because it's impossible to know whether the mere presence of the other animal was causing the EEG/physiological reaction that the goat was experiencing. Please clarify if this was incorrect. If it is the case, it might be a dealbreaker because there's no true baseline.

Response: Authors really appreciate the observation about baseline measurement raised by the reviewer. To get more clarity on the experimental design, we have added Fig 1. Goats were housed together in a lairage overnight before the start of the experiment. The control value was measured in the absence of animals at the slaughter point. The main objective of the present study was to assess the emotional stress/ animal reactions during exposure to slaughter ambient, thus two values (control-animal alone at slaughter point, and treatment stage values during exposure to the slaughtering process) were recorded and compared.

Comment: L154 - was it 5-7 minutes or 4-6 mins at noted on L133?

Response: The parameters were recorded for 5-7 min (1 min before S goat arrived + 4-6 min exposing slaughter process, till S goat was dead).

Comment: L173 - how much of each goat's recording was excluded due to artifact? How much usable data was obtained for each?

Response: Out of the recording under the Control phase (animal not visualizing the slaughter), approximately 35-45 seconds and during the treatment phase a usable data of approximately 150-180s were obtained after excluding artifacts. 

Comment:  Fig 2 - what do the black boxes on the screen represent? An epoch length? If so, add to caption/note on the figure.

Response: Added in the Figure note

Comment: I suggest moving L193-209 below the results, because they are more general statements. It would make more sense to read the results and their interpreting statements first, and then go to the more general context.

Response: Thank you very much for the valuable suggestion. We have edited it accordingly.  

Comment: L211-215 - suggest moving below Table 1 for the same reason.

Response: Thank you very much for the valuable suggestion. We have edited it accordingly.  

Comment: Table 1 - what does RMS stand for in the caption?

Response: RMS-root mean square value; It has been corrected and replaced with variables.

Comment: Table 1 - the note about superscripts is unnecessary because that is obvious from the p-values, since there are only two groups. Usually, the superscripts are only necessary when 3+groups are being compared.

Response: Thank you very much for the valuable observation. We have edited it accordingly and removed superscripts.

Comment: L238 - what do the delta waves represent? The others were all explained but not this one.

Response:A description added as- Higher delta waves were recorded in goats during slaughter as compared to the corresponding value at the farm [30]. These waves are associated with brain’s default mode network [41].

Comment: L245 - why were the gamma waves not reported for this sample? Especially when they are discussed in the Discussion as being important?

Response: Thank you very much for the valuable suggestion. We will keep it in mind and will evaluate gamma waves in our future studies.

Comment: L247 - I would avoid using the term 'slaughter empathy' because we don't know that this is what's going on. It looks like the animals are experiencing high levels of stress, but this could be emotional contagion, a cousin of empathy which is well-established in animals, but is not believed to be a type of cognitive empathy, as slaughter empathy would be.

Response: Edited as- to the heightened emotional status of goats arising due to exposure to the act of slaughter. Further, we have added this aspect of further research needed in this direction to confirm it in the section- limitation and future direction.

Comment: L258 - 'HERE' is at the end of this sentence.

Response: typo error- Deleted.

Comment: L330 - where are the limitations and future directions? What about the practical applications? Just saying that future research with a larger sample needs to be done isn't sufficient. There are other limitations that should be noted, and there are plenty of future directions that could be mentioned. Also, what does it mean for real life goats in livestock industries? Should they be protected from witnessing slaughter of other goats? Is this realistic?

Response: The limitation and future directions have been revised as-

`              This study highlighted the issue of emotional stress in goats upon exposure to slaughter ambient. There is a need to take further studies on evaluating its effect on other common and established stress biomarkers such as stress hormones. Its overall impact on meat proteomics and meat quality should be assessed.

As in the present study, all three senses (auditory, olfactory, and optic) were used to mimic the common practice in some places. A further study on which sense has more/ no effect could be useful, so to provide accurate and practical recommendations for such situations. 

There are some practical challenges while conducting such studies due to potential ethical issues. As the present study correlated the higher electric activity cerebrocortical neurons and physiological parameters, with the emotional stress during the slaughter of goats. There is a need to take more in-depth studies to confirm slaughter empathy, which if established could have wide implications in the goat meat industry such as the requirement of slaughter out of sight of a conspecific. Interestingly, the same is recommended for Halal slaughter management.

Such data will be valuable in increasing awareness among common people and also sensitize the person involved in the meat industry, thereby improving animal welfare standards. However, there is a need to take a study on emotional stress and slaughter empathy in goats with a higher sample size to get more insights into this aspect. 

Reviewer 4 Report

Dear authors:

I consider that this is a valuable document because the assessment of the physiological and electroencephalographic responses to measurement the emotions during slaughter is an actual and important topic in order to provide welfare to the animals until the last part of their lives, and as you mentioned it is becoming imperative to study this aspect of non-human animals with appropriate research methodology. In addition to this, through these changes detected in the EEG, the empathy capacity of the animals can be inferred and how witnessing the sacrifice of others can cause stress and physiological alterations. Please review the attached file for particular comments.

Author Response

The authors are thankful to the anonymous reviewer for his critical comments and observations. These comments have helped us in improving the quality of the manuscript. We have edited the manuscript accordingly. Further, it is certified that all the issues raised by the reviewer have been incorporated into the revised manuscript. All the changes were marked in RED color text.

Comment: I consider that this is a valuable document because the assessment of the physiological and electroencephalographic responses to measurement the emotions during slaughter is an actual and important topic in order to provide welfare to the animals until the last part of their lives, and as you mentioned it is becoming imperative to study this aspect of non-human animals with appropriate research methodology. In addition to this, through these changes detected in the EEG, the empathy capacity of the animals can be inferred and how witnessing the sacrifice of others can cause stress and physiological alterations.

Response: Thank you so much for your encouraging and positive observation.

Comment: Lines 37-59: This is a very interesting paragraph, I suggest that you please review these articles to complete it: Lezama-Garcfa, K. et al. Facial expressions and emotions in domestic dogs. CABI 2019, 1 No. 028 https:ljwww.researchgate.net/publication/332393645 Facial expressions and emotions in domestic animals CABI England 2019 and Mota-Rojas D, et al. Current Advances in Assessment of Dog's Emotions, Facial Expressions, and Their Use for Clinical Recognition of Pain. Animals (Basel). 2021 Nov 22;11(11):3334. https:ljdoi.org/10.3390/anillll3334.

Response: Thank you for referring high-quality articles. These articles proved very helpful for us while revising the manuscript.  

Comment: Line 42: Please write "Ethology" with capital letter.

Response: Corrected

Comment: Line 49: Please erase the double space between "[8]." and "In".

Response: Corrected

Comment: Line 54: It is correct to say "prey" or maybe it would be better to write "predator"?

Response: Authors appreciate the valuable suggestion. We have replaced prey with the appropriate word predator.

Comment: Lines 81-89: I consider that it could be interesting to add the fact that the ECG is a technology that is not available to everyone, so this could be a limitation.

Response: Edited and limitation added.

Comment: Line 104: Please put the missing parentheses after Malaysia.

Response: Edited.

Comment: Line 106: Please put the missing" after 40.7.

Response: Added.

Comment: Line 110: Could you please specify in this part which physiological parameters were evaluated? You mention it in the results section, but here you don't.

Response: Added as (heart rate, rectal temperature, breathing rate, normal/abnormal movement, and normal/abnormal activity)

Comment: Lines 193-200: I considered that these lines would be better in the introduction section than in the results/discussion section.

Response: These section has been shifted to after result on the suggestion of the Reviewer 3.

Comment: Line 266: Please erase the extra space between "conditions" and "[33]"

Response: Edited.

Comment: Line 280: Please write "Kaka" with capital letter.

Response: Corrected.

Comment: Line 469: Please erase the point.

Response: Edited

Comment: References: Please correct the list of references according to the format requested by the journal, in the same way unify the font and size.

Response: References section has been format as per standard format of Animals.

Round 2

Reviewer 2 Report

The manuscript have been improved by the recommendation of the reviwers.

For the better appearance, the resolution of the figures must be increased!

The manuscript can be accepted for publishing after a few minor changes (see attached).

Author Response

The authors are thankful to the anonymous reviewer for his positive comments and valuable observations. All the changes were marked in RED color text.

General comment: The manuscript have been improved by the recommendation of the reviwers.

Response: Thank you so much.

Comment: For a better appearance, the resolution of the figures must be increased!

Response: The resolution of figures has been improved (for fig 2-600 dpi and for fig 3 a and s b- 2000 dpi).

Comment: The manuscript can be accepted for publishing after a few minor changes (see attached).

Response: Fig is changed to figure throughout the manuscript.

Reviewer 3 Report

This manuscript is much improved, and I am happy to accept it for publication. It is useful to know that the goats were kept together in lairage up until the moment that the E-goats were removed for their control measures. This makes it clearer that it was the slaughter, per se, rather than the presence of another goat, which caused the changes to EEG and other measures.

I just have a few minor recommendations:

1. I suggest removing L91-94 which the authors added based on my previous review. That comment was not meant to be added into the ms itself, but merely raised a point that the claims must be cited. Since the authors cited the claims, that is sufficient.

2. Throughout the ms, the authors write the 'slaughter ambient'. The more appropriate term would be 'slaughter environment'. I suggest changing that throughout.

3. Table 2 - remove the reference to the superscripts in the note. The superscripts themselves have been removed but the note is still there.

Author Response

The authors are thankful to the anonymous reviewer for his positive observations.  All the changes were marked in RED color text.

General comment: This manuscript is much improved, and I am happy to accept it for publication. It is useful to know that the goats were kept together in lairage up until the moment that the E-goats were removed for their control measures. This makes it clearer that it was the slaughter, per se, rather than the presence of another goat, which caused the changes to EEG and other measures.

Response: Thank you so much for your positive and encouraging observations.

Comment: I suggest removing L91-94 which the authors added based on my previous review. That comment was not meant to be added into the ms itself, but merely raised a point that the claims must be cited. Since the authors cited the claims, that is sufficient.

Response: Deleted.

Comment: Throughout the ms, the authors write the 'slaughter ambient'. The more appropriate term would be 'slaughter environment'. I suggest changing that throughout.

Response: The word slaughter ambient is replaced by slaughter environment.

Comment: Table 2 - remove the reference to the superscripts in the note. The superscripts themselves have been removed but the note is still there.

Response: The description of superscript in the footnote has been deleted
